| 1  | Lightning Data Analysis of the CMA Network in China                                    |
|----|----------------------------------------------------------------------------------------|
| 2  | Feng Li <sup>1</sup> , Lei Wu, and Yan Li                                              |
| 3  | Meteorological Observation Center, China Meteorological Administration, Beijing,       |
| 4  | China                                                                                  |
| 5  | Corresponding author address: Feng Li, Operation Monitoring Division of                |
| 6  | Meteorological Observation Center of CMA, No.46, Zhongguancun Nandajie of              |
| 7  | Haidian District, Beijing, China. E-mail: liflif04@cma.gov.cn                          |
| 8  |                                                                                        |
| 9  | Abstract                                                                               |
| 10 | Based on analysis and evaluation of the 2009 to 2013 national lightning monitoring     |
| 11 | data, the average lightning detection station distance is approximately 170 kilometers |
| 12 | in China, and the average operational availability (AO) exceeds 90%. Lightning         |
| 13 | detection systems use a hybrid location method of direction finding (DF) and the time  |
| 14 | difference of arrival (TOA). The stations use four localization algorithms, including  |
| 15 | two-station mixing, three-station mixing, four-station mixing and two-station          |
| 16 | amplitude. Among them, the four-station method has the highest positioning accuracy,   |
| 17 | i.e., close to 50%. The statistical results show that lightning occurrences in China   |
| 18 | have increased, especially negative cloud-to-ground (-CG) flashes because positive     |
| 19 | cloud-to-ground (+CG) flashes account for only 5.1% of the total. In china, most of    |
| 20 | lightning currents range are in the -60~+60kA, Lightning current between -10~+10kA     |

| 21 | only accounted for small proportion of 0.6%. The average intensity of positive flashes     |  |
|----|--------------------------------------------------------------------------------------------|--|
| 22 | is 64.2kA, the average intensity of negative flashes is -40.28kA. The average +CC          |  |
| 23 | intensity is higher than that of -CG flashes, which is consistent with statistical results |  |
| 24 | of other Lightning detection network. Lightning frequency has obvious regional             |  |
| 25 | differences across the country; the high density lightning area is mainly distributed in   |  |
| 26 | south-central China, the south-central Yangtze River region and the eastern part of        |  |
| 27 | southwestern China. Seasonal variation in lightning activity is well defined, with few     |  |
| 28 | lightning occurrences in winter and a gradual but significant increase from spring to      |  |
| 29 | autumn in the middle and lower reaches of the Yangtze River to the north, south and        |  |
| 30 | southwest. The ratio of positive to negative flashes is highest in winter.                 |  |

Key words: lightning detection; localization algorithm; statistical features

## 32 1. Introduction

Lightning is a discharge phenomenon that commonly occurs in the development 33 phase of strong cumulonimbus clouds, and it is often accompanied by strong wind 34 gusts and rain and occasionally hail and tornadoes (Chen Weimin. 2006; Xu Xiaofeng 35 36 et al.2003). The electric current of lightning is generally tens of thousands to hundreds of thousands of amperes, and the discharge process takes less than 60 microseconds 37 (Qie Xiushu et al. 2013). The strong current, high temperature, violent wave, 38 electromagnetic field change, strong electromagnetic radiation and other properties of 39 lightning can cause fatal injuries. Lightning affects aerospace (Nie Ying et al. 2008), 40 communication (Wang ShunXiang. 2013), power (Hu XianNiu. 2003), forests (Wu 41

ShuSen et al. 2010), buildings (Hou AnXiao. 2014), national defense and the economy.
Statistics show that there are, on average, tens of thousands of lightning-related
incidents, more than 800 lightning casualties, and billions of Yuan in direct economic
losses each year in China.

China is located in temperate and subtropical regions; therefore, thunderstorm 46 activity is frequent. The 21 provincial capital cities experience more than 50 47 48 thunderstorm days each year, with as many as 134 days in some cities. The casualties, 49 economic losses and social effects caused by lightning are becoming increasingly 50 serious (China Meteorological Administration. 2008). Therefore, there is an urgent need to effectively monitor the occurrence, development and evolution of lightning. 51 The thunder and lightning monitoring and location systems produced at the end of the 52 53 20th century and beginning of the 21th century can detect lightning discharges more accurately, providing powerful data support to thunder and lightning monitoring, 54 protection and research (Lin Jian et al. 2008). 55

In order to monitor and prevent lightning events, since late twentieth Century, a 56 57 number of lightning monitoring network were established in the world, such as the United States National Lightning Detection Network (NLDN) (Cummins K L etal 58 1998a), European lightning detection network (LINET) (Betz H-D,2009)and the 59 global lightning location network (WWLLN)(Dowden RL etal,2002), the network 60 61 constantly were updated technology, improved localization algorithm, the detection efficiency continuously to improve (Rakov V A, 2013), which play an important role 62 in the research of lightning technology and lightning monitoring and warning 63

operation.

| 65 | In 2004, China began constructing a national lightning detection network that          |  |
|----|----------------------------------------------------------------------------------------|--|
| 66 | includes unified comprehensive data processing and positioning calculation. Based or   |  |
| 67 | the 2009 to 2013 national lightning monitoring data, this paper analyzes the operation |  |
| 68 | of the Chinese lightning detection network and explores the temporal and spatial       |  |
| 69 | distribution of lightning, which is expected to provide a reference for domestic       |  |
| 70 | lightning monitoring and forecasting.                                                  |  |

# 71 2. Lightning Detection Network

### 72 2.1 Network Operation Stability

Construction of China's lightning detection network began in the early 21<sup>st</sup> 73 century and was initiated by the Chinese Academy of Sciences and the China 74 Meteorological Administration (CMA). The CMA has more than 170 deployments, 75 while the Chinese Academy of Sciences has approximately 50. After 2008, most of 76 77 the lightning detection equipment was under the unified management of the CMA. Thus, the national lightning detection network formed and began operation. The 78 number of national lightning detection network observation stations reached 275 at 79 80 the end of 2009, with 265 stations being built by the CMA and only 10 stations by other ministries. From 2010 to 2013, the network continued to expand and increased 81 by 10% per year. The number of stations reached 347 at the end of 2013, and the 82 detection coverage increased by 30% compared to 2008. The average distance 83 84 between stations is approximately 170 kilometers, approximately 300 kilometers in western China and 150 kilometers in eastern China. The network station layout is 85

| 86  | shown in Figure 1. The detection accuracy is approximately 500 meters.                                                                    |  |
|-----|-------------------------------------------------------------------------------------------------------------------------------------------|--|
| 87  | Figure 1 about here                                                                                                                       |  |
| 88  | The CMA observation center evaluates the operation ability of the lightning                                                               |  |
| 89  | detection network every year. The operation availability (AO) is used as an evaluation                                                    |  |
| 90  | index. The specific algorithm is the sum of the time that all equipment was in normal                                                     |  |
| 91  | operating or suspicious conditions (the device self-deviation or crystal deviation value                                                  |  |
| 92  | is greater than the specified value, but it is still working to get data) divided by the                                                  |  |
| 93  | total the equipment operation time. The index reflects the stability of the overall                                                       |  |
| 94  | operation of the observation network.                                                                                                     |  |
| 95  | $\mathbf{AO} = \frac{Normal \ status \ running \ times + suspicious \ status \ running \ time}{total \ running \ time} \times 100\%  (1)$ |  |
| 96  | Among them, the total running time includes the normal running time, suspicious                                                           |  |
| 97  | state running time and equipment failure time. But it is also different from the full                                                     |  |
| 98  | year, the full year is equal to the sum of the total running time and maintenance time                                                    |  |
| 99  | (According to the regulation, each equipment needs to be shut down for a certain                                                          |  |
| 100 | period of time to carry out routine maintenance to prevent the failure).                                                                  |  |
| 101 | The results show that China's annual average operational availability of the                                                              |  |
| 102 | lightning network was 89.68%, 91.19%, 95.39%, 96.30%, and 94.41% in 2009, 2010,                                                           |  |
| 103 | 2011, 2012, and 2013, respectively, which shows that the overall stability of the                                                         |  |
| 104 | lightning observation network is Credible and satisfactory in China.                                                                      |  |
| 105 |                                                                                                                                           |  |
| 106 | 2.2 Lightning Location Method                                                                                                             |  |

The national lightning observation network primarily uses the ADTD model

lightning detector, which is based on the VLF/LF time difference direction hybrid 108 109 positioning technique. The device was developed by the Chinese Academy of Space Science and Application Research Center and was certified by the CMA in 2003. The 110 detector can simultaneously detect the cloud-to-ground lightning electromagnetic field 111 112 azimuth and the time of arrival. Then, an appropriate positioning algorithm can be chosen according to the number of stations that actually detect a lightning 113 114 electromagnetic wave. Four types of localization algorithms, namely two-station 115 hybrid, three-station hybrid, four-station hybrid and two-station amplitude (referred to 116 as M1, M2, M3, and M4), are used to determine the location of lightning according to the site distribution and detected data. 117

The M1, M2, and M3 algorithms are hybrid location methods that 118 comprehensively utilize direction finding (DF) and the time difference of arrival 119 (TOA). The principle is that each lightning detector can detect the hit back azimuth 120 and the time of return stroke radiation of electromagnetic waves that reach the 121 detection station. If two detectors receive the electromagnetic wave signal, the 122 123 lightning position can be calculated using a hyperbolic function of the TOA and the two azimuth angles. If three detectors receive the electromagnetic wave signal, the 124 time difference method is used in the double solution region; in the non-dual solution 125 region, the double solution is obtained using the time difference method. False 126 127 solutions are removed from the double solution using the direction-finding method. If four or more detectors receive the wave, the TOA least squares method is used to 128 calculate the best lightning position. 129

The multi-station network (more than four stations) mainly uses the TOA data to 130 131 locate lightning. Then, the TOA localization results are used for systematic error correction of the DF data to improve the positioning accuracy of the two-station 132 and three-station hybrid methods. The TOA and DF hybrid lightning location 133 system can ensure positioning results for networks with fewer detectors with high 134 positioning accuracy, which is a more practical lightning monitoring and positioning 135 136 system. According to domestic and international research, its general positioning 137 accuracy is between a few hundred meters to 2-3 kilometers, for example, U.S. 138 National Lightning Detection Network(NLDN) location error is about 308m(Nag et al.2011), LINET location error is 100~200m(Betz et al, 2009), World Wide Lightning 139 Location Network(WWLLN) location error was estimated to be 4-5km (Abarca et al. 140 2010). Recently, Wang Yu et al. (2015) study on Beijing Lightning NETwork (BLNET) 141 142 and point out its location error was less than 200 m within the network and 3 km at the range of 100 km outside the network. The M3 method has the highest efficiency in 143 actual operation (Shao Liang-qi,2007). It can not only ensure that a small number of 144 145 detection stations to participate in the positioning of the calculation, save computing resources and time, but also to ensure a high positioning accuracy, is a more practical 146 lightning monitoring and positioning method. Its frequency of use is also high, which 147 can be seen in the following analysis. 148

The amplitude method is mainly used to detect cloud-to-ground flashes and near-distance in-cloud flash. Its main principle is that according to the attenuation relationship between the intensity and the distance of the lightning radiation field, a

standard intensity value is obtained to determine the approximate distance of lightning.

Its relative error is mainly determined by the dispersion and propagation error of

lightning intensity (Zhang W J et al. 2009).

Figure 2 shows the usage frequency for the four lightning positioning algorithms 155 156 (M1, M2, M3, and M4) from 2009 to 2013. The M3 algorithm is used most often, accounting for nearly 50% of lightning detection results, and its usage has 157 158 continuously increased since 2009. M1 was used approximately 30% of the time, and 159 M2 was used approximately 20% of the time. M4 was used least often and was 160 decommissioned in 2012-2013. The results show that 40%-50% of lightning strikes can be measured using M3 in China. This result also reflects the high sensitivity of the 161 ADTD detector; in the event of a lightning strike, multiple adjacent stations can 162 simultaneously capture the corresponding electromagnetic wave signal. Therefore, the 163 positioning precision is also high. After 2012, M4, which has a lower detection 164 accuracy, was eliminated; this outcome also verifies that the detection ability of 165 China's lightning network is constantly improving. 166

#### Figure 2 about here

Figure 3 shows the cumulative frequency distributions of the detection algorithms from 2009 to 2013. The M3 algorithm has the highest frequency, especially in Sichuan, Yunnan, Guizhou, Hubei, the Yangtze and Huai River Basins, regions south of the Yangtze River and southern China, i.e., 6000 to 10000. Figure 3 also shows that the detector density is reasonable. Moreover, thunder and lightning events that can be detected in these regions using the multi-detector approaches. The

- M1 algorithm is primarily used in the Sichuan Basin, western Sichuan Plateau, the 175 border between the Yunnan and Guizhou provinces, the eastern mountain area south 176 of the Yangtze River, and the eastern and southern hilly and mountainous areas of 177 southern China, suggesting that the detection environment is poor in these areas, and 178 the lightning signal tends to be detected by only a few stations.

Furthermore, the M2 algorithm is used less often than M1 and is concentrated in the 181 Sichuan Basin and Guangdong's Zhujiang River delta. M4 has the lowest usage 182 frequency, i.e., less than 1000, and is concentrated west of Guizhou, in Chongqing, and in the southern part of the region south of the Yangtze River. According to the 183 principle of lightning detection and local convective weather and climate 184 characteristics, it can be assumed that more cloud flashes occur in these regions, 185 which are easily detected by the two-amplitude method, although its overall frequency 186 is much less than the number of cloud-to-ground flashes (Yu Min, et al 2015). Overall, 187 the detection algorithm distribution is consistent with lightning locations, and the M3 188 189 algorithm (multi-station method, high accuracy, and high sensitivity) is generally used in areas prone to lightning, while the M4 algorithm is used less often. Additionally, 190 the various detection algorithms are mainly concentrated in eastern China and are 191 rarely used in western regions and Inner Mongolia. This conclusion may be related to 192 193 climate or the lack of detectors in these areas.

Figure 3 about here

### 195 **3. Lightning Statistics**

## 196 3.1 Temporal Changes in Lightning Features

Occurrence number of detected lightning increased each year from 2009 to 2013 in China (Figure 4). This increase is, on the one hand, related to improvement in 198 lightning detection equipment and the monitoring network and, on the other hand, 199 200 may be related to climate change in China. According to monitoring statistics, there were 49772855 (cloud-to-ground) lightning events from 2009 to 2013 in China; 201 202 47245362 of these were negative lightning strikes, accounting for 94.9% of the total. 203 There were 589149 positive lightning strikes, accounting for 5.1% of the total. This 204 finding is consist