# Peer review of "Key words: lightning detection; localization algorithm; statistical features"

_Atmospheric Measurement Techniques, 2016_

## Referee Comment (RC1) · Anonymous Referee #2 · 24 Jun 2017

The paper has been improved with this revision, especially in terms of clarity and explanations. The use of references has been optimized, the figures are clearer. The responses to the comments are very relevant and satisfactory. There are only a few points remaining.

From a technical perspective, the detection of lightning mainly depends on the baseline of stations and the threshold to define a lightning event. The analysis of spatial lightning distribution only makes sense when the resolution of detection efficiencies is homogeneous, i.e. the baseline of sensors and threshold are similar. There is no description of how lightning events are detected and extracted and there is no detailed description of the algorithm for the four presented methods. In addition, the distribution of sensors is not homogeneous and some regions are much denser than other areas, and there is also no station in Tibet. I think these missing elements should be included in the discussion.

[Figure]

The LF and VLF radio signal of CG lightning can propagate over a thousand kilometers or more. The baseline of this network is only 170 km. In my opinion, it should be possible to use more than 4 sensors to determine lightning locations. This reviewer finds it confusing that the authors still use M1, M2 and M4 as they also confirm that using more stations results in better locations in this paper. Further, there is a large section discussing the use of these four methods (Section 2.2), but it is unclear what the scientific meaning of the method 'usage frequency' is. I propose the authors clarify why these methods are used and what the scientific reason of comparing them is.

Some results in this paper, such as, that thunderstorms normally occur in the summer and that positive lightning is easier to trigger in winter thunderstorms, were presented before [e.g. Rakov, V. A., and M. A. Uman, 2003, Chapter 2 & 5). The results in this paper provide a description of the literature but don't present novel results. Overall, I would highly recommend that the authors include further analysis to produce more substantial evidence. For example, the correlation between lightning occurrences and some meteorological and climate information, the spatial distribution of positive lightning, because positive lightning normally occurs close to tall objects or close to objects of moderate height located on mountain tops.

The newly added content about lightning current is brilliant. I would like to suggest to add some description about the lightning current calculation algorithm rather than just providing a result.

---

## Author Comment (AC1) · 10 Jul 2017

First of all, thank you very much for the comments of the reviewer. To sum up, reviewer mainly have 3 questions and suggestions, as following, authors answer one by one. 2.Q1. reviewer recommend adding detailed descriptions of lightning location and intensity algorithms in the manuscript. PS: Four lightning location algorithms and lightning intensity algorithms discussed in this paper have been published in detail in the literature "principles and techniques of lightning detectionïijĹMa. 2015ïijĽ".Furthermore, there are too many content descriptions and formulas in the algorithm, which is not suitable for detailed description in this article. If you are interested, you can refer to the literature(Ma.2015.Below we will provide some reference content, see Figure 1-3).

The main purpose of this paper is to analyze the application of these algorithms in the National Lightning network and the application of lightning data. In order to evaluate

the improvement of operation capability of lightning network.The algorithm itself is not the focus of this article.

Q2ïijŽreviewer wondered what's purposes and scientific reasons for analysis and comparison of lightning location algorithms 'using frequency' in this manuscripts

PSïijŽThis is a good question. As we all know, the quality of the lightning detection network depends on the efficiency and accuracy of the lightning location. However, it is very difficult to confirm the efficiency and accuracy of lightning location.However, it is generally accepted that both the hybrid method and the multi-station method are better than the two-station method, the magnetic direction method and the amplitude method in the positioning efficiency and accuracy. Due to the geographical conditions, the number of base stations and network layout, CMA lightning network can not be all unified use multi-station positioning algorithm. Our strategy is to automatically select the location algorithm in the business software based on the number of detectors detected by one of the lightning signals.From the point of view of spatial distribution, where the higher the frequency of mult-station method is, the more reasonable the base station is, the higher the quality of lightning dataãĂĆFrom the point of view of time variation, the higher frequency is selected by the multi-station method, the higher the detection ability of the whole lightning network, and the higher the quality of the lightning data, andÂăviceÂăversa. The above is the scientific reason and purpose of analyzing and comparing the frequency of using the lightning location algorithm in this paper.This also is a method to evaluate the performance of CMA lightning detection network in this paper.

Q3ïijŽReviewer recommend more analysis of lightning, especially the evidence that lightning is more likely to occur in winter, and suggest an analysis of the relationship between lightning and some climatic information. QS:That's a very good suggestion. Study on Lightning climatology, scholars including China scholars, has done a lot of analysis, including the relationship between lightning and ground temperature, water vapor, aerosol concentration, weather system and geographical conditions, also reveals the characteristics and causes of lightning climate change. According to the suggestion of reviewers, we will add analysis on the correlation between China's lightning changes and air temperature, atmospheric tropospheric layer height and atmospheric water content, and to reveals the reasons for its abnormal climatology. Thank you for your expert guidance. Specific content is added to this manuscript or another research paper.

Thanks again for the reviewers' comments and suggestion.

Please also note the supplement to this comment:
https://www.atmos-meas-tech-discuss.net/amt-2016-380/amt-2016-380-AC1-supplement.pdf

雷电监测原理与技术

马启明 编著

科学出版社

**Fig. 1.** algorithm reference book

目前比较实用的闪电定位技术。

它的定位原理是:每个探测站既探测回击发生的方位角,又探测回击辐射的电磁脉冲波形峰点到达的精确时间。当有两个探测站接收到数据时,采用一条时差双曲线和两个测向量的混合算法计算位置(图 3.24);当有三个探测站接收到数据时,在非双解区域,采用时差算法,在双解区域,先采用时差算法得出双解,后利用测向数据剔除双解中的假解(图 3.25);当有四个及四个以上探测站接收到数据时,采用时差最小二乘算法定位计算。四站定位误差如图 3.26 所示。

[Figure]

Fig. 2. Location algorithm extract examples

**Fig. 3.** Location algorithm extract examples

---

## Referee Comment (RC2) · Anonymous Referee #3 · 12 Jul 2017

The manuscript introduces a new lightning detection network for China and presents first analysis of the respective dataset. This is of interest for the scientific community and generally matches the scope of AMT. Before publication in AMT, however, major revisions are necessary.

General comments:

1. What is actually new?

It is often not clear what part of the paper contains new innovations developed by the author. Generally, the authors have to point this out clearly, and provide detailed descriptions for new aspects. Whereever instruments/algorithms/datasets etc. have been already introduced somewhere else, this has to be stated clearly as well, with appropriate references. The respective descriptions within the current paper might then be just summarized, but any modification of existing algorithms etc. has to be

clarified.

2. What is the scope of the manuscript?

According to the title, it is on "lightning data analysis", which is quite vague. From the manuscript, I see two major aspects: a) description of the network, the lightning retrieval algorithms, accuracies etc. b) first results of lightning distribution over China.

a) description of the network/algorithm. As far as I understand, this paper provides the first description of the CMA LDN and thus will serve as reference for upcoming studies. If this is the case, there are several important aspects missing:

- the description of the network should be given in more detail (physical pricinple, history of number of stations etc.)

- what are future plans? (maintenance, extension)

- will the data be available for the scientific community?

But if I am wrong here and the description of the network and the algorithm is given elsewhere, this has to be clarified and referenced appropriately. In addition, the respective paragraph might be shortened.

b) first results of lightning distribution over China This part is quite short. If this aspect is meant to be the main focus of this study, it has to be extended considerably. At least a comparison to the LIS climatology with focus on spatial patterns, seasonal and diurnal cycles should be added.

see https://lightning.nsstc.nasa.gov/data/data_lis-otd-climatology.html

Please clarify the main scope of the manuscript, which in any case needs major revisions of the text (adding references + shorten text for existing stuff, provide more details for the new results). The title should point this out as well.

3. References

Only few references are given to prior work, and many of the given references miss a doi and can not be found, or are in Chinese language. In particular the general introduction to the topic of lightning completely misses the pioneering works of the last century.

Detailed comments:

Abstract: Please carefully revise the abstract after clarifying the overall scope of the manuscript and revising the paper respectively. Try to make the abstract concise, short, but still give one sentence on the scope of the paper in the beginning (before giving any numbers).

Abstract (line 17-18): What does 50% accuracy mean? Lightning occurrence has not necessarily increased, might be just caused by increased detection efficiency (as stated later in the text!)

Introduction: Please provide appropriate references, i.e. the pioneering studies on the respective topic, or easily accessible, english review articles on lightning in general.

Sections 2.1/2.2: Add appropriate references to previous work. What is the physical principle of the lightning detection (frequency, antennas)? Please discuss similarities and differences to other networks like NLDN, WWLLN, LINET). What about IC flashes? Are they detected as well? Can they be discriminated from CG? Discuss! Provide references for M1/M2/M3 methods.

Section 3: Which algorithm was used for the results shown in section 3?

Summary: Revise according to general comment 2.

Figs. 1, 3, 5, 6: What is the reason for the insertion in the lower right corner? This should be skipped or clarified.

---

## Author Comment (AC2) · 3 Aug 2017

Thank you very much for the reviewers' comments. The questions and suggestions are very good, which will help us to make further improvements. Now we will answer all the questions one by one.

Comment1: what is actually new and innovation of this paper? PS: This is a good question. So far, no scholars (including Chinese scholars) have carried out a comprehensive, systematic introduction, analysis and evaluation of the CMA lightning detection network. This paper mainly introduces the development history of CMA LDN, and analyzes the operational capability of CMA lightning network by using the index used in the service. This paper also introduces lightning location algorithms that can be automatic selected to use by CMA LDN business software, according to the number of detectors which detected same one the lightning signals. Change of various algorithms

use frequency is analyzed to prove the network performance and positioning accuracy improvement from another perspective. Based on the analysis of the CMA LDN, this paper uses the data of 5 years to analyze the temporal and spatial distribution characteristics and climate characteristics of lightning activities in China, these results can be used to compare with abroad similar networks, which should help foreign counterparts to understand China LDN. Otherwise, it would be a pity that there is no recent China's lightning detection network information in the world literatures database.

Comment2: What is the scope of manuscript? PS: First, I'm sorry. Title of the manuscript is modified according to another reviewer's suggestion. Maybe it's not very suitable, but we have accepted it.

As reviewer say, the main content of this manuscript has two parts. The first part mainly introduces the basic information, development history and operational capability of CMA lightning detection network, including location algorithm brief description and business application. Referring to the reviewer's comment, we add some basic information about lightning location equipment (line 66-72) in the introduction section. The history of the number of detection stations has been introduced in section 2.1. About future plans, the manuscript has added some information in the line 73-78, the following diagram gives us the basis of the layout design of CMA lightning sites and the distribution plan of the site in 2020. Owing to the length of the article, we are not going to add it to the manuscript. Please see the specific content in revised paper. With regard to data reliability, we think the reliability of lightning data has been proved by analysis of section 1 and 2 in the paper.

Fig.1 Left is theoretical distribution map of detection efficiency, right is CMA LDN station distribution plan map (blue point is existed station, red point is planned station, total station number is 599) .

The main content of the second part, is to analyze Chinese lightning activity spatial and temporal distribution and climate change characteristics by using 5-year data collected

by CMA LDN. The reliable data reveals the fact of Chinese lightning event and inter-annual variation possible reason. These results can be used to compare with the existing results of other lightning nets. Limited to the length of the article, this article does not make a comparison. ButïijŇaccording to reviewer's comment , we have added an analysis related to other climate variables in section 3.1.

Comment 3ïijŽAbout references PS: According to the journal requirements and expert comments, we have supplemented all foreign reference DOI, but for the Chinese literature, I regret that, due to differences in file system functionality, DOI cannot be provided.

Some responses to detailed comments Comment4: abstract need be revised PS: Thank you very much for your comments. We have made a careful revision of the abstract according to your opinion, and more clearly defined the purpose and scope of this study.

Comment5: "what does 50% accuracy mean?(line 17-18)" PS: As we all know, the quality of the lightning detection network depends on the efficiency and accuracy of the lightning location. However, it is very difficult to confirm the efficiency and accuracy of lightning location. Generally, it is accepted that both the hybrid method and the multi-station method are better than the two-station method, the magnetic direction method and the amplitude method in the positioning efficiency and accuracy. Due to the geographical conditions, the number of base stations and network layout, CMA LDN cannot be all unified use multi-station positioning algorithm. Our strategy is to automatically select the location algorithm by the business software based on the number of detectors detected same one lightning signals. From the point of view of spatial distribution, where the higher the usage frequency of multi-station method is, the more reasonable the detection station location is, the higher the quality of lightning dataãĂĆFrom the point of view of time variation, the higher multi-station method usage frequency is, the higher the detection ability of the whole lightning network, and the higher the quality of the lightning data, and vice versa. According to statistics, in practical applications, M3

usage frequency has increased year by year, the frequency of use is close to 50% by 2013.Therefore, we believe that the detection accuracy of the whole CMA LDN is improving continuously. Thus, the sentence "50% accuracy"(line 17-18) is not expressed accurately and has been modified in the abstract.

Comment6: "Which algorithm was used for the results shown in section 3?" PS: According to explanation in Comment5, I believe reviewers have already understood that the lightning data used in section 3 is the result of various algorithms. The location algorithm of each lightning event is chosen automatically according to the number of stations.

Comment7: Figs. 1, 3, 5, 6: What is the reason for the insertion in the lower right corner? This should be skipped or clarified. PS: According to the Chinese publication requirement and regulations, the South China Sea Islands must be printed in the lower right corner of China map. This is already an international rule, and scholars familiar with Chinese scientific research have accepted it. We hope you understand it and thank you for reminding me.

Acknowledgments Firstly, thanks to comments of reviewers, we have made a great deal of amendments and supplements to the full text. I would like to express my heartfelt thanks to you.

Because of the length and scope of paper, it is impossible for us to involve all scientific and technological problems of Chinese lightning. Therefore, we cannot meet the different concerns of all reviewers in this manuscript, we express our apologies. This also shows that international communications in the field of China's lightning research and business work is too little, and there is not enough reference and literature to provide foreign scholars. Next, we will continue to improve CMA LDN and timely share the development of China's lightning research and business development with foreign counterparts. Finally, we would like to express our heartfelt thanks to reviewers.

Please also note the supplement to this comment:
https://www.atmos-meas-tech-discuss.net/amt-2016-380/amt-2016-380-AC2-
supplement.pdf

---

## Author Comment (AC3) · 29 Aug 2017

First of all, thank you very much for the comments. We have revised some content in manuscript according to the comments. The followings are point-by-point replies to the comments. In order to distinguish, our replies and modifications are used in red text.

Comment1: "From a technical perspective, the detection of lightning mainly depends on the base-line of stations and the threshold to define a lightning event. The analysis of spatial lightning distribution only makes sense when the resolution of detection efficiencies is homogeneous, i.e. the baseline of sensors and threshold are similar. There is no description of how lightning events are detected and extracted and there is no detailed description of the algorithm for the four presented methods. In addition, the distribution of sensors is not homogeneous and some regions are much denser than other areas, and there is also no station in Tibet. I think these missing elements should

be included in the discussion."

PS: Thank you for reviewer comments. Q(1): Indeed, as reviewer say, the analysis of lightning data must be based on the homogeneous of the detection network and detector, i.e. threshold is similar. CMA LDN uses a unified model detector ADTD, to achieve homogeneous of sensor and threshold. Because the network inhomogeneous densities or inhomogeneous distribution of sensors, we use an automatic selection strategy for lightning location algorithms. For example, in the Tibetan Plateau and the surrounding site sparse area, the system can choose the "two station" algorithm, as long as a lightning weak signal can be received by two sensors, the lightning is detected and can be positioned. In the plains, more detectors involved in the detection and location, the system can choose "three station" or "four station" algorithms, which has higher accuracy. According to the lightning detection principle, different baseline may cause the difference in detection efficiency, but the strategy has been reduced the error to a minimum, under realistic conditions, this way is the most practical and reliable.

Q (2): This manuscript has introduced CMA LDN distribution and constructer, "2.2 Lightning Location Method"give lightning location method description, and we add detector ADTD information in introduction section. We think we have described how lightning was detected and extracted. Sorry, we don't know what else need to be added? Whether we need to give the working principle of the lightning detector in this manuscript? We don't think this is the focus of this job.

Q(3): Four lightning location algorithms are introduced in section "2.2 Lightning Location Method", and their more detailed description have been published in the literature "principles and techniques of lightning detectionïïjĹMa. 2015ïïjĽ". There are too many content descriptions and formulas about four algorithms, which is not suitable for detailed description in this article. If you are interested, you can refer to the literature (Ma,Q.M 2015). We have added literature information in references part. Below we will provide some reference content, see Figure 1-3). The main purpose of this paper is to

analyze the application of these algorithms in the National Lightning network and the application of lightning data. And to evaluate the improvement of operation capability of lightning network by algorithms usage frequency variation. The algorithm itself is not the focus of this article.

Q(4): These discuss have been added in Summary and Conclusions (line 315-327) and other part (line 142-143).

(fig1-3 are Lightning location and current algorithms reference book and some algorithms examples)

Comment 2: The LF and VLF radio signal of CG lightning can propagate over a thousand kilometers or more. The baseline of this network is only 170 km. In my opinion, it should be possible to use more than 4 sensors to determine lightning locations. This reviewer finds it confusing that the authors still use M1, M2 and M4 as they also confirm that using more stations results in better locations in this paper. Further, there is a large section discussing the use of these four methods (Section 2.2), but it is unclear what the scientific meaning of the method 'usage frequency' is. I propose the authors clarify why these methods are used and what the scientific reason of comparing them is.

PS: This is a good question. Reviewer wondered what's purposes and scientific reasons for analysis and comparison of lightning location algorithms 'using frequency' in this manuscripts As we all know, the quality of the lightning detection network depends on the efficiency and accuracy of the lightning location. However, it is very difficult to confirm the efficiency and accuracy of lightning location. It is generally accepted that both the hybrid method and the multi-station method are better than the two-station method, the magnetic direction method and the amplitude method in the positioning efficiency and accuracy. Furthermore, the more sensors are involved in the localization algorithm, the more accurate the location is. However, by 2013, our multi-station positioning algorithm was most applied to the 4 sensors, and more than 4 sensors have

not been applied to the multi-station positioning algorithm. Therefore, the multi-station method mentioned in this paper is the four station method. Due to the geographical conditions, the number of base stations and network layout, CMA lightning network cannot be all unified use multi-station positioning algorithm. Our strategy is to automatically select the location algorithm in the business software based on the number of detectors detected by one of the lightning signals. From the point of view of spatial distribution, where the higher the frequency of mult-station method is, the more reasonable the base station is, the higher the quality of lightning data. From the point of view of time variation, the higher frequency is selected by the multi-station method, the higher the detection ability of the whole lightning network, and the higher the quality of the lightning data, andÂăviceÂăversa. The above is the scientific reason and purpose of analyzing and comparing the frequency of using the lightning location algorithm in this paper. This also is a method to evaluate the performance of CMA lightning detection network in this manuscript.

Comment 3: Some results in this paper, such as, that thunderstorms normally occur in the summer and that positive lightning is easier to trigger in winter thunderstorms, were presented before [e.g. Rakov, V. A., and M. A. Uman, 2003, Chapter 2 5). The results in this paper provide a description of the literature but don't present novel results. Overall, I would highly recommend that the authors include further analysis to produce more substantial evidence. For example, the correlation between lightning occurrences and some meteorological and climate information, the spatial distribution of positive lightning, because positive lightning normally occurs close to tall objects or close to objects of moderate height located on mountain tops.

QS: That's a very good suggestion. Study on Lightning climatology, scholars including China scholars, has done a lot of analysis, including the relationship between lightning and ground temperature, water vapor, aerosol concentration, weather system and geographical conditions, also reveals the characteristics and causes of lightning climate change. According to the suggestion of reviewers, we add analysis on the correlation

between China's lightning changes and air temperature, atmospheric troposphere layer height and atmospheric water content, and to reveals the reasons for its abnormal climatology (line 213-233). According to reviewer suggestions, we analyzed the spatial distribution and changes of positive lightning in winter and summer. Winter (December as a representative,fig.5), positive lightning mainly distribute in southern Anhui, northern Jiangxi, North Central Guangxi, Yunnan and Western Xinjiang, these areas are basically mountainous, above 1500 meters above sea level. And summer (especially in June,fig4), positive lightning widely distribute in China's central and eastern regions, whether plain or mountain. According to the location of lightning, further statistics (fig6) shows that winter positive lightning is the most prone to in the 1000-2000 meters height mountains, accounted for 42.4The above content has been added to the manuscript (line 294-311), please review.

Fig.7-1-2 Positive lightning spatial distribution in Jun (left)and Dec (right)

Fig7-3 Positive lightning vertical distribution proportion with altitude in winter and summer

Comment 4: The newly added content about lightning current is brilliant. I would like to suggest to add some description about the lightning current calculation algorithm rather than just providing a result. PS: Thank you very much for your suggestion. Because the lightning current algorithm is also more complex, it will take too much space to describe here. The scope of this article mainly analyzes the lightning data of China, we does not want to involve more lightning detection technology. Lightning current algorithm has been published, if you are interested, the specific content can refer to reference (Ma, Q.M, 2015).

Below is an overview of the algorithms. Lightning current formula: (sorry, formula is missing here, formula please see PDF file) , This is a fitting formula, used of artificial lightning and lightning positioning system synchronous observation, obtained two sets of independent observation data, and then the two sets of data fitting. Among them,

SNF expressed signal normalization factor. RNSS represents normalized to 100Km signal strength, and SS represents the original electric or magnetic field signal strength. In CMA LDN ADTD lightning positioning system, $\beta$ is set to 1 (assuming the only signal measured is the radiation field), $\lambda$ is set to infinity, SNF=0.392. RNSS' is averaging the normalized signal intensity RNSS of all station.

Thanks again for the reviewer's comments and suggestion.

Please also note the supplement to this comment:
https://www.atmos-meas-tech-discuss.net/amt-2016-380/amt-2016-380-AC3-supplement.pdf

雷电监测原理与技术

马启明 编著

科学出版社

**Fig. 1.** Lightning location and current algorithms reference book

目前比较实用的闪电定位技术。

它的定位原理是:每个探测站既探测回击发生的方位角,又探测回击辐射的电磁脉冲波形峰点到达的精确时间。当有两个探测站接收到数据时,采用一条时差双曲线和两个测向量的混合算法计算位置(图 3.24);当有三个探测站接收到数据时,在非双解区域,采用时差算法,在双解区域,先采用时差算法得出双解,后利用测向数据剔除双解中的假解(图 3.25);当有四个及四个以上探测站接收到数据时,采用时差最小二乘算法定位计算。四站定位误差如图 3.26 所示。

[Figure]

图 3.24 两站 IMPACT 系统的定位示意图

图 3.25 三站 IMPACT 系统的双解区域定位示意图

**Fig. 2.** location algorithms examples printscreen

总结:在处理两站闪电定位时(所对应 $\theta_A$、$\theta_B$、……测置,计算出基线附近区域的参数(所对应 $\theta_A$、$\theta_B$、……采用振幅对比法计算闪电位置,闪电所发生的大致区域,在基线附近的区域定位方法计算。但有时,在处理基线附近区在基线附近的区域以外,采用基础两站定位方法计算。但有时,在处理基线附近区域内的闪电时,由于场强误差的影响,两站数据无法交汇。

域内的闪电时,由于场强误差的影响,场强法算法模型

3) 三站和三站以上磁方向闪电定位算法模型

当使用三站或者三站以上的站进行闪电定位时,由于测量误差的影响,交汇结果是一个球面三角形或者球面多边形。理论表明:当探测角 $\theta_i$ 的误差分布是随机误差分布时,闪电最有可能发生的点是使得各探头理论探测角 $\alpha_i$ 之差的平方和为最小时,误差三角形或者误差多边形内的点。

设闪电点的坐标为 $P(\sigma,\Omega)$,闪电监测网共有个 $N_d$ 探头,其中第 $i$ 个探头的坐标为 $D_i(\sigma_i,\Omega_i)$,测得的闪电方位角为 $\theta_i$,场强为 $E_i$,并设理论闪电方位角为 $\alpha_i$,另外,还假定 $\theta_i$ 的误差为随机误差分布(事实上还有系统误差,并且系统误差较随机误差大并不满足高斯分布),设为 $e_i$,并根据前面的电波传播理论,将权定义为 $W_i = E_i$,则对应第 $i$ 个探头的误差方程为

$$e_i = \alpha_i - \theta_i, \quad i = 1,2,\cdots,N_d \tag{3.33}$$

定义 $IR = \sum\limits_{i=1}^{N_d} E_i \,(\alpha_i - \theta_i)^2$,由最小二乘法原理,闪电最佳位置该满足下列方程:

$$\partial IR/\partial\sigma = 0, \quad \partial IR/\partial\Omega = 0$$

即:

$$\partial IR/\partial\sigma = 2\sum\limits_{i=1}^{N_d} E_i(\alpha_i - \theta_i)\partial\alpha_i/\partial\sigma = 0 \tag{3.34}$$

$$\partial IR/\partial\Omega = 2\sum\limits_{i=1}^{N_d} E_i(\alpha_i - \theta_i)\partial\alpha_i/\partial\Omega = 0 \tag{3.35}$$

另外,如图 3.14 所示,在球面三角形 $NPD$ 中,分别利用正余弦定理有

$$\sin\alpha_i = \frac{\cos\sigma\sin(\Omega - \Omega_i)}{\sin\delta_i} \tag{3.36}$$

图 3.14  球面三角形 $NBP$

**Fig. 3.** location algorithms examples printscreen

**Mean Total Positive Lightning during Jun(2009-2013)**

[Figure]

**Fig. 4.** Positive lightning spatial distribution in Jun

Mean Total Positive Lightning during Dec(2009-2013)

**Fig. 5.** Positive lightning spatial distribution in Dec

Positive lightning vertical distribution proportion with altitude in winter and summer

**Fig. 6.** Positive lightning vertical distribution proportion with altitude in winter and summer

---

## Author Comment (AC4) · 29 Aug 2017

Thank you very much for the reviewers' comments. The questions and suggestions are very good, which will help us to make further improvements. Now we will answer all the questions one by one.

Comment1: "what is actually new and innovation of this paper?"

PS: This is a good question. So far, no scholars (including Chinese scholars) have carried out a comprehensive, systematic introduction, analysis and evaluation of the CMA lightning detection network. This paper mainly introduces the development history of CMA LDN, and analyzes the operational capability of CMA lightning network by using the index used in the service. This paper also introduces lightning location algorithms that can be automatic selected to use by CMA LDN business software, according to the number of detectors which detected same one the lightning signals. Change of

various algorithms use frequency is analyzed to prove the network performance and positioning accuracy improvement from another perspective. Based on the analysis of the CMA LDN, this paper uses the data of 5 years to analyze the temporal and spatial distribution characteristics and climate characteristics of lightning activities in China, these results can be used to compare with abroad similar networks, which should help foreign counterparts to understand China LDN. Otherwise, it would be a pity that there is no recent China's lightning detection network information in the world literatures database.

Comment2: "What is the scope of manuscript?"

PS: First, I'm sorry. Title of the manuscript is modified according to another reviewer's suggestion. Maybe it's not very suitable, but we have accepted it.

As reviewer say, the main content of this manuscript has two parts. a) The first part mainly introduces the basic information, development history and operational capability of CMA lightning detection network, including location algorithm brief description and business application. Referring to the reviewer's comment, we add some basic information about lightning location equipment (line 66-72) in the introduction section. The history of the number of detection stations has been introduced in section 2.1. About future plans, the manuscript has added some information in the line 73-78, the following diagram gives us the basis of the layout design of CMA lightning sites and the distribution plan of the site in 2020. Owing to the length of the article, we are not going to add it to the manuscript. Please see the specific content in revised paper. With regard to data reliability, we think the reliability of lightning data has been proved by analysis of section 1 and 2 in the paper. About algorithm, we added its reference in line 142-143, but I'm sorry, it is in Chinese.

Fig.1 is theoretical distribution map of detection efficiency. fig.2 is CMA LDN station distribution plan map (blue point is existed station, red point is planned station, total station number is 599) .

b) The main content of the second part is to analyze Chinese lightning activity spatial and temporal distribution and climate change characteristics by using 5-year data collected by CMA LDN. The reliable data reveals the fact of Chinese lightning event and inter-annual variation possible reason. Here, according to two reviewers comment, we have added and analyzed the relationship between the number of China's lightning and the ground temperature, precipitation and CAPE, and obtained the climatic reasons for the annual change of lightning. These results can be used to compare with the existing results of other lightning nets. Limited to the length of the article, this article does not make a comparison. ButïijŇaccording to reviewer's comment , we have added an analysis related to other climate variables in section 3.1.

Comment 3: References "Only few references are given to prior work, and many of the given references miss a doi and cannot be found, or are in Chinese language. In particular the general introduction to the topic of lightning completely misses the pioneering works of the last century."

PS: According to the journal requirements and reviewer comments, we have supplemented all foreign reference DOI, but for the Chinese literature, I regret that, due to differences in file system functionality, DOI cannot be provided. About some pioneering works of the last century, many papers have often introduced, and this article no longer describes them. Thank you for your suggestion.

Some responses to detailed comments Comment4: "Abstract: Please carefully revise the abstract after clarifying the overall scope of the manuscript and revising the paper respectively. Try to make the abstract concise, short, but still give one sentence on the scope of the paper in the beginning (before giving any numbers)."

PS: Thank you very much for your comments. We have made a careful revision of the abstract according to your opinion, and more clearly defined the purpose and scope of this study.

Comment5: "Abstract (line 17-18): What does 50% accuracy mean? Lightning occurrence has not necessarily increased, might be just caused by increased detection efficiency (as stated later in the text!)"

PS: As we all know, the quality of the lightning detection network depends on the efficiency and accuracy of the lightning location. However, it is very difficult to confirm the efficiency and accuracy of lightning location. Generally, it is accepted that both the hybrid method and the multi-station method are better than the two-station method, the magnetic direction method and the amplitude method in the positioning efficiency and accuracy. Due to the geographical conditions, the number of base stations and network layout, CMA LDN cannot be all unified use multi-station positioning algorithm. Our strategy is to automatically select the location algorithm by the business software based on the number of detectors detected same one lightning signals. From the point of view of spatial distribution, where the higher the usage frequency of multi-station method is, the more reasonable the detection station location is, the higher the quality of lightning data. From the point of view of time variation, the higher multi-station method usage frequency is, the higher the detection ability of the whole lightning network, and the higher the quality of the lightning data, and vice versa. According to statistics, in practical applications, M3 usage frequency has increased year by year, the frequency of use is close to 50% by 2013.Therefore, we believe that the detection accuracy of the whole CMA LDN is improving continuously. Thus, the sentence "50% accuracy"(line 17-18) is not expressed accurately and has been modified in the abstract. According to the analysis of the annual change relationship between lightning number and climatic factors, the number of lightning in China has increased in 2009-2013 years, but it has not always been increasing, and it has annual decreasing variation in 2012. This phenomenon cannot be caused by the increase of detection efficiency. However, this lightning number increase must also be partly caused by an increase in detection efficiency.

Comment6: "Introduction: Please provide appropriate references, i.e. the pioneering studies on the respective topic, or easily accessible, english review articles on lightning

in general. Sections 2.1/2.2: Add appropriate references to previous work. What is the physical principle of the lightning detection (frequency, antennas)? Please discuss similarities and differences to other networks like NLDN, WWLLN, LINET). What about IC flashes? Are they detected as well? Can they be discriminated from CG? Discuss! Provide references for M1/M2/M3 methods."

PS: First, thank you for suggestion. According to the subject and the length of the present content, we will not add too many references, some classical literature, many articles are often introduced, and we do not no longer repeat them. Sections 2.1/2.2: we add location algorithm reference. We add Chinese lightning detector ADTD information in introduction section (line 64-70), including physical principle of the lightning detection (frequency, antennas). I'm sorry, we don't discuss similarities and differences to other networks like NLDN, WWLLN, LINET in this manuscript. We only provide CMA LDN information, which can be used to compare with other networks by other interested researchers. According to ADTD mechanical structure and physical principle, most of IC strokes weak signal are filtered out. Of course, lightning information also contain a small amount of cloud flash pulses information, but the results can not accurately distinguish IC flash and CG flash, they also cannot be discriminated from CG. M1/M2/M3 methods detailed algorithms are described in the reference book "principles and techniques of lightning monitoring" (Ma,Q.M 2015). (Ma, Q.M, 2015: Principle and technology of lightning monitoring. Science Press, Beijing, 101-181.)

Comment7: "Which algorithm was used for the results shown in section 3?"

PS: According to explanation in Comment5, I believe reviewers have already understood that the lightning data used in section 3 is the result of various algorithms. The location algorithm of each lightning event is chosen automatically according to the number of stations.

Comment8: "Summary: revise according to general comment 2"

Ps: summary has been revised.

Please also note the supplement to this comment:
https://www.atmos-meas-tech-discuss.net/amt-2016-380/amt-2016-380-AC4-supplement.pdf

––––––––––––––––––––––––––––––––––

[Figure]

[Figure]

**Fig. 1.** Fig.1 theoretical distribution map of detection efficiency

[Figure]

**Fig. 2.** fig.2 CMA LDN station distribution plan map